# Comparison of the Serion IgM ELISA and Microscopic Agglutination Test for diagnosis of *Leptospira* spp. infections in sera from different geographical origins and estimation of *Leptospira* seroprevalence in the Wiwa indigenous population from Colombia

Anou Dreyfus[1,2,3]*, Marie-Thérèse Ruf[1,2], Marga Goris[4], Sven Poppert[1,2], Anne Mayer-Scholl[5], Nadine Loosli[1,2], Nadja S. Bier[5], Daniel H. Paris[1,2], Tshokey Tshokey[6,7], John Stenos[8], Eliharintsoa Rajaonarimirana[3], Gustavo Concha[9], Jorge Orozco[10], Johana Colorado[11], Andrés Aristizábal[12], Juan C. Dib[12,13], Simone Kann[14]

1 Department of Medicine, Swiss Tropical and Public Health Institute, Basel, Switzerland, 2 University of Basel, Basel, Switzerland, 3 Epidemiology and Clinical Research Unit, Institut Pasteur de Madagascar, Antananarivo, Madagascar, 4 Department of Medical Microbiology and Infection Prevention, OIE and National Collaborating Centre for Reference and Research on Leptospirosis, Amsterdam University Medical Centre, Amsterdam, the Netherlands, 5 Department of Biological Safety, German Federal Institute for Risk Assessment, Berlin, Germany, 6 Department of Pathology and Laboratory Medicine, Jigme Dorji Wangchuck National Referral Hospital, Thimphu, Bhutan, 7 Faculty of Postgraduate Medicine, Khesar Gyalpo University of Medical Sciences of Bhutan, Thimphu, Bhutan, 8 Australian Rickettsial Reference Laboratory, University Hospital Geelong, Geelong, Australia, 9 Organización Wiwa Yugumaiun Bunkuanarua Tairona (OWBYT) Valledupar, Colombia, 10 La Secretaría de Salud Departamental de la Gobernación de Cesar, Valledupar, Colombia, 11 Laboratorio de Salud Pública de la Secretaría de Salud Departamental, Valledupar, Colombia, 12 Fundación Salud Para El Trópico (FSPT), Santa Marta, Colombia, 13 Department of Medicine, Fundación Universidad del Norte, Barranquilla, Colombia, 14 Medical Mission Institute, Wuerzburg, Germany

* anoudreyfus@outlook.com, anou.dreyfus@pasteur.mg

## Abstract

Leptospirosis is among the most important zoonotic diseases in (sub-)tropical countries. The research objective was to evaluate the accuracy of the Serion IgM ELISA EST125M against the Microscopic Agglutination Test (MAT = imperfect reference test); to assess its ability to diagnose acute leptospirosis infections and to detect previous exposure to leptospires in an endemic setting. In addition, to estimate the overall *Leptospira* spp. seroprevalence in the Wiwa indigenous population in North-East Colombia. We analysed serum samples from confirmed leptospirosis patients from the Netherlands (N = 14), blood donor sera from Switzerland (N = 20), and sera from a cross-sectional study in Colombia (N = 321). All leptospirosis ELISA-positive, and a random of negative samples from Colombia were tested by the MAT for confirmation. The ELISA performed with a sensitivity of 100% (95% CI 77% - 100%) and a specificity of 100% (95% CI 83% - 100%) based on MAT confirmed *Leptospira* spp. positive and negative samples. In the cross-sectional study in Colombia, the ELISA performed with a sensitivity of 100% (95% CI 2–100%) and a specificity of 21% (95% CI 15–28%). Assuming a 5% *Leptospira* spp. seroprevalence in this population,

**Data Availability Statement:** Data is available at: https://www.kaggle.com/datasets/anousati/dreyfus-colombia-elisa-mat-results.

**Funding:** The authors D.P., A.D., S.P. received funding from the R. Geigy Foundation: https://en.geigystiftung.ch. The authors S. K., A.D.received funding from the Else Kröner Fresenius-Stiftung: (EKFS), 2016_HA 190; https://www.ekfs.de/en The funders had no role in study design, data collection and analysis, decision to publish, or preparation of the manuscript.

**Competing interests:** The authors have declared that no competing interests exist.

the positive predictive value was 6% and the negative predictive value 100%. The *Leptospira* spp. seroprevalence in the Wiwas tested by the ELISA was 39%; however, by MAT only 0.3%. The ELISA is suitable to diagnose leptospirosis in acutely ill patients in Europe several days after onset of disease. For cross-sectional studies it is not recommended due to its low specificity. Despite the evidence of a high leptospirosis prevalence in other study areas and populations in Colombia, the Wiwa do not seem to be highly exposed to *Leptospira* spp.. Nevertheless, leptospirosis should be considered and tested in patients presenting with febrile illness.

## Author summary

Leptospirosis is among the most important zoonotic diseases in (sub-)tropical countries. The correct diagnosis of leptospirosis is very important to take a medical or public health decision. Therefore, we tested a serological test (ELISA) for its ability to correctly diagnose a negative sample as truly negative and a positive sample as truly positive. We tested the ELISA with European acute leptospirosis confirmed positive and negative samples and compared results with another serological test (microscopic agglutination test), which is the recognized reference test. Further, the ELISA was assessed for its ability to detect previous exposure to leptospires in serum samples from the indigenous Wiwa population from Colombia, where leptospirosis is expected to be endemic.

The ELISA performed very well with sera from patients with acute leptospirosis, however had difficulties to diagnose negative samples as truly negative in the Colombian field samples; hence unexposed persons were falsely diagnosed to be positive. Therefore, we recommend using the ELISA to detect acute leptospirosis several days after onset of illness in a non-endemic environment, but are not convinced of its usefulness to screen a population for previous *Leptospira* spp. exposure.

## 1. Introduction

Leptospirosis is a worldwide prevalent zoonotic disease and among the most widely spread endemic diseases in subtropical and tropical countries. Transmission follows exposure to urine or tissues of infected mammals, either through direct contact or via contaminated water or soil. *Leptospira* are spirochete bacteria that comprise currently 68 species and more than 250 different serovars [1–4]. A large range of mammalian hosts (i.e. rodents, cattle, dogs, pigs, etc.) carry host adapted serovars in their renal tubules and excrete them into the environment over months to years. A warm, humid environment with humans and animals in close vicinity of common water sources is the ideal setting for leptospires to become endemic. Humans may develop severe or even life-threatening illness following infection as accidental hosts [1,5,6].

The World Health Organization (WHO) estimated the worldwide annual incidence of leptospirosis at 1.03 million cases and 58,900 deaths [7] with the highest burden occurring in resource-poor tropical countries, including countries of Latin America [8] and Asia [9].

In this study we tested human sera for the presence of anti-leptospiral antibodies from three countries: the Netherlands, Switzerland, and Colombia with an IgM ELISA (Serion ELISA classic *Leptospira* IgM, Institut Virion\Serion GmbH, EST125M), in the text referred to as "ELISA") and the reference Microscopic Agglutination Test (MAT).

In the Netherlands, leptospirosis is endemic and has been a mandatory reportable disease since 1928. During 1925–2008, the average incidence was 0.25 cases/100,000 population [10]. In 2014 there was a marked increase to 0.57 cases/100,000 population [11], which more or less remained at this level up to 2020. The reported incidence likely represents the more severe end of the clinical spectrum for leptospirosis, because mild forms of this disease more commonly go unrecognized [1].

In Switzerland a mandatory reporting system for human leptospirosis is not in place and therefore the leptospirosis incidence and burden in Switzerland are unknown. However, if leptospirosis was prevalent at high levels, at least severe cases would have been recognized, as in general, infectious diseases with severe symptoms are followed up diagnostically. Sporadically, human case reports are published, such as the cluster of leptospirosis cases among river surfers in Northern Switzerland [12], or acute autochthonic leptospirosis in patients in Southern Switzerland [13].

In Colombia, human leptospirosis is under mandatory notification rule since 2007 [14]. In 2010, an incidence of 2.9 cases and in 2012 of 2.2 per 100,000 inhabitants was reported [15]. However, leptospirosis is most likely underreported due to unspecific symptoms, lack of awareness and reliable diagnostic tests.

Leptospires induce a serovar-specific immune response consisting of a cellular (type 1) and humoral (type 2) mediated immunity. Sero-conversion may occur after 2–10 days after onset of disease, therefore, serological antibody detecting tests will only become sensitive several days after onset of symptoms. IgM cass antibodies usually appear earlier than IgG class antibodies, and remain for months or years at a low detectable titre. IgG antibodies may not be detected at all, or for only a short period, or persist for several years [5,16]. The duration of detectable antibodies in humans after natural infection with *Leptospira* spp. varied substantially within and between studies, with sero-positive persons becoming sero-negative between 6 and 60 months after infection. Detection methods for leptospires are either direct, identifying *Leptospira* spp. antigen or genomic substances, or indirect, identifying host antibodies. Diagnosis of acute leptospirosis is increasingly made using PCR, as leptospires are present in blood and cerebrospinal fluid in the first week after onset of symptoms and then in urine until the third week [6]. PCR for detection of pathogenic leptospires is sensitive and specific in the acute phase, but not in the convalescent and cannot distinguish between serogroups or serovars, which is relevant for attributing a source in epidemiological studies. However, molecular methods to identify the genome species and to discriminate between infecting strains from DNA extracted directly from the primary sample [17,18], are increasingly important in epidemiological studies [19] and are continuously improved [20]. For human and veterinary serological studies, at present the gold standard remains the Microscopic Agglutination Test (MAT), which is not specific for any particular class of antibody, but differentiates between serogroups and serovars [5]. To diagnose an acute infection with *Leptospira*, either a MAT titre ≥400 (probable case) or a fourfold rise in titre between sera taken five to 10 days apart (confirmed case) is recommended. A single MAT titre is mainly useful to detect previous exposure in cross-sectional studies (MAT titre ≥100). The MAT has a poor sensitivity (22.6%) during the acute phase of disease [21,22]. As shown in a study by Goris et al., the sensitivity and specificity of the MAT improves with convalescent serum of patients, who found a sensitivity of 88% and a specificity of 98% [23]. The MAT can be adapted to any epidemiological situation and region, as its panel of serogroups and serovars may be modified for the local setting, if endemic serogroups are known. Unfortunately, the MAT is very laborious, cost intense and subjective. Therefore, it is usually only implemented in reference laboratories. Other laboratories prefer screening sera for anti-leptospiral antibodies by ELISA, subsequently confirming the positive samples by MAT in the reference laboratory. Serological assays like

ELISAs and Rapid Diagnostic Tests (RDTs) share disadvantages of low sensitivity in the early phase of disease, however, in convalescent serum, accuracy becomes much better [24,25]. In addition, the performance of an ELISA or rapid diagnostic test may be hampered by the existence of different endemic *Leptospira* spp. serogroups, non-specific reactions or by other previous or current infections in the tested patient/study participant. It is therefore important to evaluate these tests for a specific epidemiological "setting".

The objectives of our study were to evaluate the sensitivity and specificity of the commercially available Serion ELISA classic *Leptospira* IgM using the MAT as imperfect reference test with patient sera from confirmed leptospirosis cases, blood donor sera, with sera from patients with other infections than leptospirosis (all European) and sera from a cross-sectional study in the indigenous population called "Wiwas" of the Sierra Nevada de Santa Marta, North-east of Colombia.

In addition, we aimed at estimating the overall *Leptospira* spp. seroprevalence in the Wiwa indigenous tribe in Colombia.

## 2. Materials and methods

### 2.1 Study settings

**2.1.1 Patient sera from the Netherlands who suffered from acute leptospirosis (group A).**   These sera were used to test the ability of the ELISA to diagnose clinical leptospirosis. Paired sera (including an admission and follow-up sample) from patients, suspected to suffer from leptospirosis, were submitted to the OIE ("World Organisation for Animal Health") and National Collaborating Centre for Reference and Research on Leptospirosis (NLR), which is located at the Academic Medical Center, Department of Medical Microbiology and Infection Prevention, University of Amsterdam and were analyzed by qPCR and/or culture and/or MAT for confirmation of leptospirosis (see case definition). General practitioners and consulting clinicians suspecting leptospirosis routinely send clinical specimens to the NRL for laboratory evaluation, where 99% of suspected cases are investigated.

Of these patients, 14 convalescent/follow-up samples were taken, 22 to 50 (median 32) days post onset of illness and sent to the Swiss Tropical and Public Health Institute, Basel, Switzerland (Swiss TPH) for the ELISA evaluation. The infecting serogroups were Icterohaemorrhagiae, Sejroe, Shermani, Bataviae or Pomona with a geometric mean of the corresponding (highest) MAT titre of 640 (range 160–5120). See **Supporting Information** for detailed characterizaition of these samples.

**2.1.2 Sera from healthy blood donors from Switzerland (group B).**   Twenty sera from healthy blood donors collected in Switzerland available at the Swiss Tropical and Public Health Institute (Swiss TPH) were included in the validation with the objective to test the ability to diagnose persons without leptospirosis diagnosis as negative. Sera were anonymized and no clinical, travel or occupational background was available. These sera were classified as originating from persons without clinical leptospirosis and most likely seronegative for *Leptospira* spp..

**2.1.3 Patient sera with a confirmed diagnosis of another infection (group C).**   In order to determine specificity and assess potential serological cross-reactivity of the ELISA, sera collected during routine diagnostics at the diagnostic center of the Swiss TPH were tested of patients with and without travel history. Included were sera reactive with antigens of different helminths, protozoan, bacterial and viral infections (see **Table 1**). The selection of these sera was based on availability and not on scientific evidence of cross-reactions between *Leptospira* spp. and these particular pathogens. These sera were classified as originating from persons without clinical leptospirosis and most likely seronegative for *Leptospira* spp..

**Table 1.  Sera from clinically ill patients with a confirmed parasitic, bacterial or viral infection tested by the Serion IgM ELISA (REF EST125M) for anti-leptospiral antibodies and partially confirmed by the Microscopic Agglutination Test (MAT).**

| Pathogen | No of samples tested | ELISA test result | MAT test result |
|---|---|---|---|
| *Toxocara canis* | 2 | Negative (2/2) | - |
| *Fasciola hepatica* | 2 | Negative (2/2) | - |
| *Echinococcus spp.* | 2 | Negative (2/2) | - |
| *Filaria spp.* | 5 | Negative (5/5) | - |
| *Schistosoma spp* | 5 | Negative (5/5) | - |
| *Trichinella spiralis* | 3 | Negative (3/3) | - |
| *Trichinella spiralis* | 2 | Negative (2/2) | Negative (2) |
| *Strongyloides stercoralis* | 2 | Negative (2/2) | - |
| *Entameoba histolytica* | 2 | Negative (2/2) | - |
| *Leishmania spp.* | 5 | Negative (5/5) | Negative (1) |
| *Trypanosoma cruzi* | 2 | Negative (2/2) | - |
| *Plasmodium spp.* | 3 | Negative (3/3) | - |
| Dengue | 5 | Negative (5/5) | - |
| *Rickettsia spp.* | 1 | Negative (1/1) | - |
| Total | 41 | | |

**2.1.4 Sera from the Wiwa population, Colombia (group D).**    These sera were used for two purposes: (i) to test the capacity of the ELISA to detect past *Leptospira* spp. exposure and (ii) to estimate the *Leptospira* spp. seroprevalence in the Wiwa population. The sera were collected during a program against Chagas Disease in July and November 2014, hence in the rainy and dry season from an indigenous tribe called Wiwa [26]. The Wiwa live in retracted areas of the Sierra Nevada de Santa Marta, north-east of Colombia. Since they usually remain inside their territory, little is known e.g. about their burden of leptospirosis. Their access to medical care is sparse and infectious diseases are very common, due to risk factors like climatic conditions, poor socioeconomic status, proximity to livestock, the lack of access to clean drinking water (river, unprotected wells) and sanitation, simple housing (mud walls, uncoated floor, palm roofs), traditional agriculture practices, etc. In total 489 serum samples from volunteers (healthy or ill) were collected, after obtaining written informed consent. Of these samples, a by

**Table 2.  Overview of origin of serum samples, diagnostic tests and laboratories involved in the evaluation of the Serion IgM ELISA (REF EST125M), which was performed at the Swiss Tropical and Public Health Institute (Swiss TPH), Basel, Switzerland.**

| Group | Origin/ laboratory | Data source/ study design | Leptospirosis status | No of samples tested by ELISA[1] | No of samples tested by MAT | Laboratory conducting MAT | Other diagnostic tests performed |
|---|---|---|---|---|---|---|---|
| A | NLR, the Netherlands | Leptospirosis patient sera from hospitals sent to the NLR | Confirmed = "true positives" | 14 | 14 | NLR | qPCR, culture or IgM In-house ELISA at NLR |
| B | Swiss TPH, Switzerland | Blood donor sera | "Supposedly" negative | 20 | 0 | not done | not specified |
| C | Swiss TPH, Switzerland | Sera of patients with confirmed infections | "Supposedly" negative | 41 | 3 | NLR | see Table 3 |
| D | MMI, Germany/ Colombia | Cross-sectional study in Colombian Wiwa population | Exposed and unexposed field samples | 321 | 156 | NLR | not specified |

[1]Serion IgM ELISA EST125M. Abbreviations: qPCR real time Polymerase Chain Reaction; MAT Microscopic Agglutination Test; ELISA Enzyme linked Immunosorbent Assay; **NLR** National Collaborating Centre for Reference and Research on Leptospirosis at the Academic Medical Centre, Department of Medical Microbiology, University of Amsterdam, the Netherlands; **MMI** Medical Mission Institute, Wuerzburg, Germany

village stratified random sample of 321 persons was drawn from the villages Tezhumake ((n = 119, 37%), Department Cesar), Ashintuwa (n = 66, 21%), Cherua (n = 61, 19%) and Seminke (n = 75, 23%) (Department La Guajira). The sera were stored at -20˚C in the Laboratorio Salud Publica in Valledupar and transported by World Courier to the Julius Maximilian University, Wuerzburg, Germany. In 2019, the samples were shipped to the Swiss TPH to be tested for leptospiral antibodies by the ELISA and thereafter to the National Collaborating Centre for Reference and Research on Leptospirosis (NLR), University of Amsterdam to be tested by MAT as confirmatory test. During all shipments the cooling chain was maintained. **Table 2** provides an overview of laboratories involved and diagnostic tests used.

## 2.2 Ethical statement

The use of the sera from patients in the Netherlands (group A) was exempted from ethical review of human subject research by the Medical Ethical Review Committee of the Academic Medical Centre, University of Amsterdam (written protocol W12_075#12.17.0092). All data have been anonymized and were not attributable to individual patients. For patient sera with a confirmed diagnosis of a viral, bacterial or parasitic infection (group C) a written permission to use the sera for test evaluations was obtained ("Unbedenklichkeitsbescheinigung": "UBE-15/22") from the Swiss ethics committee "Ethik Kommission Nordwest- und Zentralschweiz" (EKNZ). According to the EKNZ, based on the Swiss law on science in humans ("Humanforschungsgesetz"), the use of anonymized, bio banked sera (as the blood donor sera, group B) does not require an ethics agreement for the development and validation of diagnostic tests and was therefore not obtained. The cross-sectional study (group D) was performed in accordance with the principles of the Declaration of Helsinki and was approved by the Ethics Committee of Santa Marta, Colombia (Acta No 032018) for the Colombian study population. For children, formal consent was obtained from the parent or legal guardian.

## 2.3 Serological testing

To assess the diagnostic accuracy of the ELISA, we tested all sera by ELISA and thereafter all ELISA-positive and a random sample of the ELISA-negative sera by MAT for the above described study populations from Colombia. The serum collection from leptospirosis patients in the Netherlands had already been confirmed by MAT before being tested by the ELISA (**S1 File**). **Table 2** provides an overview of laboratories involved and diagnostic tests used.

**2.3.1 Enzyme linked immunosorbent assay.** For the detection of anti-leptospiral IgM antibodies the Serion ELISA classic Leptospira IgM was used (Institut Virion\Serion GmbH, EST125M) according to the manufacturer's protocol. Briefly, for the detection of IgM antibodies a pre-incubation step to absorb the rheumatoid factor with Rf-absorbens for 15 min at room temperature is necessary. For the analysis, the required amount of pre-coated wells (sample(s) + negative control + standard 1 + standard 2 + substrate blank) is placed into the strip holder. Of the appropriate sample dilution 100 μl are added to the respective well for 60 minutes followed by four washing steps and the addition of 100 μl of the ready-to-use conjugate (alkaline phosphatase) for 30 minutes and again followed by 4 washing steps. Afterwards 100 μl of the p-Nitrophenylphospate substrate is added and again incubated for 30 min. Finally, 100 μl ready-to use stop-solution is added and the OD is measured at a wavelength of 405 nm against the substrate blank measured at a wavelength of 620 nm. For the evaluation, the specified calculation excel sheet from the manufacturer was used, interpreting the OD values automatically as positive, negative or ambiguous.

**2.3.2 Microscopic agglutination test.** The presence of antibodies against specific panels of pathogenic *Leptospira* spp. was assessed by MAT according to OIE standards [21,27]. Live

cultures of *Leptospira* spp. reference strains were used (**Table 3**). The selection of the *Leptospira* test panel appropriate for the study regions is based on the published information and on the available isolated strains and serogroups at the reference laboratory from Colombia and the Netherlands and on commonly prevalent serogroups worldwide, recommended by WHO [28,29]. The sera were screened at a dilution of 1:20. Those with a positive reaction were titrated in a serial two-fold dilution to determine the end-point titre defined as the reciprocal of the highest serum dilution at which ≥ 50% of the leptospires remain agglutinated.

**2.3.3 Reproducibility and repeatability.**   In order to assess the reproducibility and repeatability the inter-assay variability was determined. For the inter-assay variability, the mean value, the standard deviation as well as the coefficient of variance was calculated for two samples, tested at eight different days.

## 2.4 Case definitions

The confirmation of suspected cases of acute leptospirosis in the Netherlands requires a four-fold rise in MAT titer between two samples taken approximately 10 days apart or a PCR and/or culture positivity and/or a leptospirosis diagnosis based on clinical symptoms combined with a MAT (titre ≥1:160). While in an endemic setting a MAT titer of ≥1:800 is required to confirm leptospirosis, in the Netherlands a MAT titre ≥1:160 is considered sufficient due to a low infection background in the Netherlands. This case definition was validated by the NLR on culture positive patients (M. Goris, personal communications) and was only used on the "Dutch sera" (group A).

For the sero-epidemiological studies from countries, where leptospirosis is considered endemic, sera with a single MAT titre of ≥ 100 were considered seropositive, i.e. indicating past exposure to leptospires [30], independent of the ELISA result.

ELISA-negative sera and those with a MAT titre <100 were defined seronegative (**Fig 1**).

## 2.5 Analysis

### 2.5.1 Sample size calculations.

*2.5.1.1 Validation of the ELISA in comparison to the MAT using sera from the cross-sectional study in the Colombian Wiwa population potentially exposed to Leptospira spp.* For the validation of the ELISA in comparison to the MAT, we calculated the required sample size to detect a two-tailed statistically significant difference between two independent proportions (P = sensitivity or specificity) [31], using epitools from https://epitools.ausvet.com.au [32]. With P1 (specificity of MAT) being 90% [23] and P2 (specificity of ELISA) 50%, the confidence level 95% and the power 80, the required sample size was 50. With P1 (sensitivity of MAT) being 85% [23] and P2 (sensitivity of ELISA) 99%, the confidence level 95% and the power 80, the required sample size was 144.

*2.5.1.2 Validation of the performance of the ELISA using confirmed leptospirosis positive and negative samples.* When assessing the ELISA with samples for which the disease status had been confirmed by several diagnostic tests and clinical exam (Dutch patient samples) or where absence of disease was highly likely (blood donor samples), we applied the method described by Hajian-Tilaki [33] using the (epiR) package in R [34] (R code in **S1 File**) to determine the sample size. We estimated the diagnostic sensitivity and specificity to be 99%, setting the disease prevalence at 40% (14/34, see Table 4), and the confidence at 95%. Further, we expected that our estimate of sensitivity or specificity was within 0.07 of the true population. To determine sensitivity and specificity of the ELISA in this setting, we estimated a sample size of 20, and 13, respectively.

*2.5.1.3 Estimation of a single proportion: the Leptospira spp. seroprevalence in the Wiwa population from Colombia.* To estimate the *Leptospira* spp. seroprevalence in the Wiwa

**Table 3. Strains, serogroups and serovars of *Leptospira* spp. used as live antigens in the Microscopic Agglutination Test (MAT) for the populations in the Netherlands (N = 14), and Colombia (N = 156).**

| Genomspecies | Serogroup | Serovar | Strains used for Colombia[1] | Strains used for the Netherlands[1] |
|---|---|---|---|---|
| *L. biflexa* | Andaman | Andamana | - | CH11 |
| *L. interrogans* | Australis | Australis | Ballico | Ballico |
| *L. interrogans* | Australis | Bratislava | - | Jez Bratislava |
| *L. interrogans* | Autumnalis | Autumnalis | Akiyami A | |
| *L. interrogans* | Autumnalis | Rachmati | - | Rachmat |
| *L. interrogans* | Bataviae | Bataviae | Swart | Swart |
| *L. interrogans* | Canicola | Canicola | Hond Utrecht IV | Hond Utrecht IV |
| *L. weilli* | Celledoni | Celledoni | - | Celledoni |
| *L. interrogans* | Hebdomadis | Hebdomadis | Hebdomadis | Hebdomadis |
| *L. interrogans* | Icterohaemorrhagiae | Copenhageni | M20 | Wijnberg |
| *L. interrogans* | Icterohaemorrhagiae | Icterohaemorrhagiae | RGA | Kantorowic |
| *L. interrogans* | Pomona | Pomona | Pomona | Pomona |
| *L. noguchii* | Pomona | Proechimys | - | 1161 U |
| *L. interrogans* | Pyrogenes | Pyrogenes | Salinem | Salinem |
| *L. interrogans* | Sejroe | Hardjo type Prajitno | Hardjoprajitno | Hardjoprajitno |
| *L. borgpetersenii* | Sejroe | Hardjo type Bovis | - | Lely 607 |
| *L. borgpetersenii* | Ballum | Ballum | Castellon 3 | Mus 127 |
| *L. borgpetersenii* | Javanica | Javanica | Veldrat Batavia 46 | - |
| *L. borgpetersenii* | Javanica | Poi | - | Poi |
| *L. interrogans* | Sejroe | Saxkoebing | - | Mus 24 |
| *L. borgpetersenii* | Sejroe | Sejroe | M84 | M 84 |
| *L. borgpetersenii* | Tarassovi | Tarassovi | Perepelitsin | Perepelitsin |
| *L. kirschneri* | Grippotyphosa | Grippotyphosa | Moskva V | Duyster and Mandemakers |
| *L. kirschneri* | Cynopteri | Cynopteri | 3522 C | 3522 C |
| *L. noguchii* | Djasiman | Huallaga | M7 | - |
| *L. santarosai* | Mini | Ruparupae | M3 | - |
| *L. borgpetersenii* | Mini | Mini | - | Sari |
| *L. noguchii* | Panama | Panama | CZ 214K | CZ 214K |
| *L. santarosai* | Sarmin | Weaveri | CZ 390 | - |
| *L. interrogans* | Sejroe | Wolffi | 3705 | - |
| *L. biflexa* | Semaranga | Patoc | Patoc I | Patoc I |
| *L. meyeri* | Semaranga | Semaranga | - | Veldrat Sem 173 |
| *L. santarosai* | Shermani | Shermani | 1342 K | 1342 K |

[1]Tested at the National Collaborating Centre for Reference and Research on Leptospirosis at the Academic Medical Centre, Department of Medical Microbiology, University of Amsterdam, the Netherlands

population, we estimated a sample size of 139 participants for a precision of 0.05, a confidence of 0.95% and an estimated apparent 10% *Leptospira* spp. seroprevalence using epitools from https://epitools.ausvet.com.au [32].

**2.5.2 Determination of the performance of the ELISA.** We used the MedCalc online tool (https://www.medcalc.org/calc/diagnostic_test.php) [35] to calculate sensitivity, specificity, positive and negative predictive values and 95% confidence intervals of the "ELISA" based on test results with sera from the Colombian study population (D) or "sera collections" (A-B) (**Table 4**).

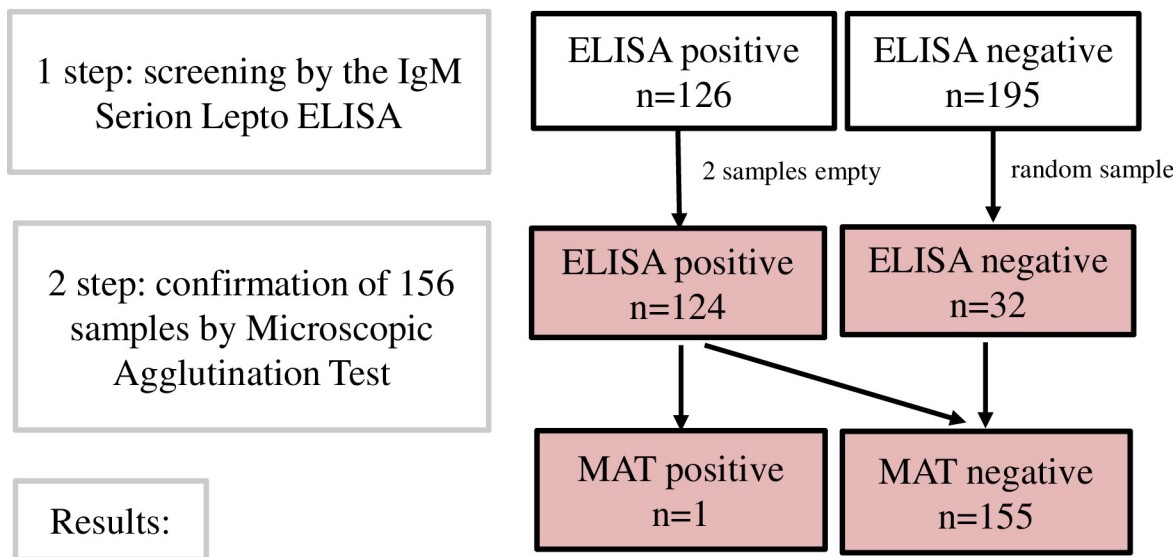

**Fig 1. The 321 sera of the Colombian study population were screened by the Serion ELISA classic *Leptospira* IgM (Institut Virion\Serion GmbH, REF EST125M).** All ELISA-positive (n = 124) and a random sample of the ELISA-negative samples (n = 32) were tested for confirmation by the Microscopic Agglutination Test (MAT = imperfect reference test) to assess the performance of the ELISA (in rose). Only one of the ELISA-positive serum samples tested *Leptospira* seropositive by the MAT, while 123 ELISA positive samples were negative in the MAT. All ELISA negative results, tested by MAT were confirmed by the MAT as negative.

**2.5.3 Estimation of *Leptospira* spp. seroprevalence in the Colombian population.** Data was recorded in Microsoft Excel and analyzed with Stata 15. Proportions of seropositive persons overall were calculated. Confidence intervals of proportions (95% CI) were calculated by the Fleiss Method [36].

# 3. Results

## 3.1 Performance of the "ELISA" assessing acute leptospirosis (group A and B)

We tested 14 sera from Dutch confirmed leptospirosis cases ("true positives") with the ELISA and all resulted with a positive ELISA test. The 20 blood donor sera ("most likely negatives") tested all negative. Hence, there were no false positive or false negative results, leading to a sensitivity of 100% (95% CI 77% - 100%) and a specificity of 100% (95% CI 83% - 100%). Therefore, the ELISA was classified/confirmed to be highly sensitive and specific when tested with European convalescent and negative sera (**Table 4**).

## 3.2 Cross-reactions with other pathogens (group C)

We then tested the ELISA for cross-reactivity with sera from 41 patients, which were clinically and diagnostically confirmed to have had either a parasitic, viral or bacterial infection but no

**Table 4. Performance of the Serion IgM ELISA (REF EST125M) tested with sera from persons from the Netherlands, Switzerland, and Colombia with different disease/exposure status compared to the results of the MAT, calculated by the MedCalc online tool (https://www.medcalc.org/calc/diagnostic_test.php).**

| Group | Origin of study population | N | ELISA pos/neg | MAT pos/neg | Sens (%) | 95% CI | Spec (%) | 95% CI | PPV % | 95% CI | NPV % | 95% CI |
|---|---|---|---|---|---|---|---|---|---|---|---|---|
| A | The Netherlands (clinical cases) and | 14 | 14/0 | 14/0 | 100.0 | 76.8-100.0 | | | NA[2] | - | - | - |
| B | Switzerland (blood donors) | 20 | 0/20 | not tested | - | - | 100.0 | 83.2-100.0 | - | - | NA[2] | - |
| C | Switzerland (patients with other infections) | 41 | 0/41 | only 3 of 41 tested – these were negative | NA[2] | NA[2] | 100.0[8] | 91.4-100.0 | NA[2] | - | NA[2] | - |
| D (i) | Colombia (cross-sectional) ELISA screening | 321[5] | 126/195 | NA[4] | NA[4] | - | NA[4] | - | NA[4] | - | NA[4] | - |
| (ii) | Confirmation by MAT | 156[5] | 124/32[7] | 1/155 | 100.0 | 2.5-100.0 | 20.6 | 14.6-27.9 | 6.2[1] | 5.8-6.7 | 100.0[1] | -[3] |

CI = Confidence interval; PPV = Positive Predictive Value; NPV = Negative Predictive Value, Sens = Sensitivity of ELISA; Spec = Specificity of ELISA; MAT Microscopic Agglutination Test; ELISA = Enzyme linked Immunosorbent Assay

[1] based on an estimated seroprevalence of 5%

[2] NA = not applicable, as only negative results (no cross-reactions) were found

[3] the MedCalc online tool did not impute a result

[4] NA = not applicable, as this row shows the ELISA screening test results

[5] we tested 321 sera by the ELISA, of which 126 (39%) were ELISA positive, and 195 (61%) were negative. Of these 321 samples, a sample of 156 were tested by MAT, including 124 ELISA-positives and a random sample of the ELISA-negatives (n = 32); only one of the ELISA-positive serum samples tested *Leptospira* seropositive by the MAT, while 123 ELISA positive samples were negative in the MAT. All ELISA negative results were confirmed by the MAT as negative

[7] These samples were actively selected to be tested by the confirmatory test

[8] the specificity, was not calculated based on MAT results, but from diagnostic test results being negative for other pathogens.

diagnosis of leptospirosis (Table 1). These tested negative by ELISA, leading to a specificity of 100% (95% CI 91–100).

Hence, the ELISA did not cross-react with antibodies against other diseases listed in Table 1.

## 3.3 Performance of the "ELISA" assessing past exposure to *Leptospira* spp. with sera from the Wiwa population in the Sierra Nevada, Colombia (group D)

We tested 321 sera by the ELISA, of which 126 (39%) were ELISA positive, and 195 (61%) negative. We sent 156 of these samples to the NRL, including all ELISA-positives (n = 124), apart from two positives due to low serum quantity and a random sample of the ELISA-negatives (n = 32) to be confirmed by MAT.

Only one of the ELISA-positive serum samples tested *Leptospira* seropositive by the MAT, while 123 ELISA positive samples were negative in the MAT (MAT titre < 100, see Table 5), and hence classified as "false positives". All ELISA negative results were confirmed by the MAT as negative, resulting with 32 "true negatives" and zero "false negatives". If we consider the MAT as the reference test, the ELISA performed in the Colombian study population with a sensitivity of 100% (95% CI 2–100%) and a specificity of 21% (95% CI 15–28%). Assuming a 5% *Leptospira* spp. seroprevalence in this population, the PPV was 6% (95% CI 6–7) and the NPV 100% (Fig 1, Table 4).

### 3.4. Reproducibility and repeatability

The inter-assay coefficient of variance over eight measurements from different days and two different samples was 8% and 10% respectively with a standard deviation of 0.1 and 0.11.

**Table 5. Distribution of Microscopic Agglutination Test (MAT) antibody titer of 156 serum samples originating from the Colombian Wiwa tribe against the *Leptospira* serovar panel listed below (strains and serogroups are listed in Table 3).** A MAT titer cut-off of ≥1:100 was classified as seropositive. Given the large serovar panel, the quantity of serum was not always sufficient to test all samples against all serovars. The number and percentage of samples not tested for a given serovar are visible in the last column.

| Serovar | | MAT Titer | | | | | |
|---|---|---|---|---|---|---|---|
| | | 0 | 20 | 40 | 80 | 160 | Not tested |
| Shermani | N (%) | 150 (96.1) | 5 (3.2) | 1 (0.6) | - | - | - |
| Patoc | N (%) | 145 (92.9) | 9 (5.8) | 1 (0.6) | 1 (0.6) | - | - |
| Wolffi | N (%) | 156 (100.0) | - | - | - | - | - |
| Hardjo | N (%) | 155 (1.0) | 1 (0.6) | - | - | - | - |
| Sejroe | N (%) | 155 (99.4) | - | - | - | - | 1 (0.6) |
| Weaveri | N (%) | 156 (100.0) | - | - | - | - | - |
| Pyrogenes | N (%) | 152 (97.4) | 2 (1.3) | 2 (1.3) | - | - | - |
| Pomona | N (%) | 154 (98.7) | 2 (1.3) | - | - | - | - |
| Panama | N (%) | 149 (95.5) | 4 (2.6) | 1 (0.6) | 2 (1.3) | - | - |
| Ruparupae | N (%) | 98 (62.8) | 1 (0.6) | - | - | - | 57 (36.5) |
| Javanica | N (%) | 98 (62.8) | - | 1.0 (0.6) | - | - | 57 (36.5) |
| Icterohaemorrhagiae | N (%) | 152 (97.4) | 4 (2.6) | - | - | - | - |
| Copenhageni | N (%) | 119 (76.2) | - | - | - | - | 37 (23.7) |
| Hebdomadis | N (%) | 154 (98.7) | 2 (1.3) | - | - | - | - |
| Grippotyphosa | N (%) | 150 (96.1) | 2 (1.3) | - | - | - | 4 (2.6) |
| Huallaga | N (%) | 95 (60.9) | 2 (1.3) | 1 (0.6) | 1 (0.6) | - | 57 (36.5) |
| Cynopteri | N (%) | 151 (96.8) | 5 (3.3) | - | - | - | - |
| Canicola | N (%) | 152 (97.4) | 3 (1.9) | 1 (0.6) | - | - | - |
| Bataviae | N (%) | 98 (62.8) | 1 (0.6) | - | - | - | 57 (36.5) |
| Castelloni | N (%) | 156 (100.0) | - | - | - | - | - |
| Autumnalis | N (%) | 115 (73.7) | - | - | - | - | 41 (26.3) |
| Australis | N (%) | 150 (96.1) | 4 (2.6) | - | 1 (0.6) | 1 (0.6) | - |

N Number of samples; % proportion of samples;—no sample in this category

### 3.5 Prevalence of leptospiral antibodies tested by ELISA and MAT with sera from the Wiwa population in the Sierra Nevada, Colombia (group D)

The seroprevalence against *Leptospira* spp. tested by the ELISA, was 39% (95% CI 35–46; 126/321). However, after confirmation of 156 samples in the MAT, seroprevalence (against any serovar) of the entire study population was much lower at 0.3% (95% CI 0.05–2; 1/319—two positive samples had insufficient serum quantity and could not be tested by MAT; therefore, the denominator was reduced from 321 to 319). The serogroup (sg) reacting in the MAT was sg Australis with a titer of 160. Several sera reacted with low MAT titers <100, however these were considered seronegative, as recommended by Levett [30]. Given the large serovar panel, the quantity of serum was not always sufficient to test all samples against all serovars. The number of samples tested by serovar are listed in **Table 5**.

## 4. Discussion

We evaluated a commercially available *Leptospira* IgM ELISA in variable epidemiological settings.

With the confirmed "leptospirosis patient sera" from the Netherlands (A) we tested the ability of the ELISA to detect clinically ill patients, and with the blood donor samples (B) the ability to diagnose most likely leptospirosis negative (i.e. without disease) persons as negative. The

ELISA always detected a *Leptospira* spp. infection in convalescent serum of ill patients and was at all times negative when serum was tested from most likely *Leptospira* spp. negative Swiss/ European population. Hence the sensitivity and specificity were high and similar to the manu- facturer's documentation of a sensitivity of 95% and a specificity of >99. A similar French study conducted by Trombert-Paolantoni et al. in 2009, revealed for the same commercial IgM ELISA using MAT confirmed negative (N = 49) and positive (N = 30) serological samples, a sensitivity of 90% (95% CI 73–98) and specificity of 82% (95% CI 68–9) (the calculation of these performance parameters were made by us, as the authors only published the total and the number of positive and negative samples for the ELISA and the MAT)[37]. The data sug- gests, that the Serion ELISA classic *Leptospira* IgM is useful to diagnose leptospirosis in a set- ting with low likelihood of previous exposure as in acutely ill Europeans several days after onset of symptoms/disease. Nevertheless, a confirmatory test is always needed/recommended to verify the result.

With the Colombian (D) cross-sectional study population we evaluated the capacity of the ELISA to detect "past exposure" to leptospires. In this Latin-American "field setting", the sensi- tivity remained high (100% (95% CI 2–100%)), however the specificity decreased strongly, being reduced to 20.6% (95% CI 15–28%) and having a positive predictive value of only 6%.

Hence, for cross-sectional studies this ELISA is not recommended, due to its low specificity. While the point estimate of the sensitivity of the ELISA was high in the Wiwa population, the very large 95% confidence interval of 2–100% indicates a large uncertainty. To correct for the high proportion of false positive results, a confirmatory test is a requirement, which represents a significant limitation and is economically not attractive, given the low PPV and the large number of false positive samples, which need confirmatory testing. The reason for those false- positive results may be due to the ELISA plates being coated with an *L*. biflexa antigen, a non- pathogenic *Leptospira* species found in the environment ubiquitously, leading to unspecific cross-reactions, as has been shown by Niloofa et al [38]. However, this cannot explain the entire problem. Given our evaluation results with the sera collections from the Netherlands and the Swiss TPH (donor sera and sera from patients with various bacterial, viral and para- sitic infections), the "false positive" ELISA results of the Colombian sera cannot be attributed to a systematic error in the implementation of the diagnostic test.

Despite the high specificity demonstrated with the sera from patients infected by other pathogens, the high burden of other infectious diseases in the Wiwa study population might have contributed to non-specific reactions leading to the low specificity. Cross-reactions were shown in a validation study conducted by Trombert-Paolantoni et al., where one of 15 sera from patients with Influenza, 6/16 with Syphilis, 2/17 infected with EBV and 2/13 patients with Borreliosis reacted with a false positive result in the same IgM ELISA[37]. Moving the positivity cutoff for a positive ELISA result towards a higher optical density (OD) would reduce sensitivity, but increase specificity, which is not recommendable for a screening test. Other studies in endemic settings, reported as well low specificity for *Leptospira* IgM ELISAs. The "Standard Diagnostics Leptospira IgM ELISA" was assessed (2001–2003) for detection of acute leptospirosis in febrile adults admitted in Vientiane, Laos. Using the cut-off suggested by the manufacturer, the assay demonstrated limited diagnostic capacity with a sensitivity of 95% and a specificity of 41% compared with the MAT [39]. Likewise, in a study in febrile patients in Thailand (2001–2002), the Panbio ELISA had, when using the cutoff value recommended by the manufacturer, a sensitivity and specificity of 91% and 55%, respectively[40]. A poor cor- relation between the MAT and an IgM ELISA was also observed by Hem et al. [41]; however, in patients with acute fever. The authors reasoned with missing serogroups in the MAT panel and with a higher sensitivity of the ELISA towards IgM antibodies. In our study, the MAT

panel was large and specifically chosen for the study area and should not have contributed to the poor correlation.

The apparent prevalence of specific anti-leptospiral antibodies in the Wiwa population from Colombia was high when tested by the ELISA (39%) and very low when tested by MAT (0.3%). We assume, that the first value most likely is an over- and the second an underestimation. A strong discrepancy of tested prevalence of *Leptospira*-specific antibodies between this Serion ELISA and the MAT was also found in a cross-sectional study conducted in Bhutan, where an apparent *Leptospira* spp. seroprevalence of 18% (95% CI 15–20, 152/864) was measured by the same IgM ELISA and 2% (95% CI 1–3, 14/864 persons) by MAT. In this study population, the same ELISA performed with a sensitivity of 86% (95% CI 57–98%) and a specificity of 63 (95% CI 57–67%), when compared to MAT [42]. The different ELISA specificities in the Colombian and Bhutanese population stress the importance of a robust validation for diagnostic tests in different populations and highlight that the positivity cutoff values may vary for different regions, settings or populations, and require adequate assessment.

It is challenging to get a precise picture on leptospirosis in Colombia, as research is concentrated on certain areas and not all published studies used recommended case definitions. A systematic literature review on the *Leptospira* spp. (sero-)prevalence in humans and animals in Colombia published between 2000 and 2012, using MAT (cut-offs not defined) and ELISA as diagnostic methods, found (sero-)prevalence to be between 6% and 35% in humans, 41% and 61% in cattle, 10% in pigs, 12% and 47% in canines, 23% in non-human primates, and between 25% and 83% in rodents [43]. A prospective study between 2012 and 2013 on undifferentiated tropical febrile illnesses (UTFI) in a tropical area of Cordoba, tested 100 fever patients for dengue, leptospirosis, hantavirus, malaria, rickettsia, brucellosis, hepatitis A and B. Leptospirosis was apparently the most common cause of UTFI with 27% testing positive by MAT. However, they did not show MAT results of all patients in the article and the titer cut-off chosen was too low to diagnose an acute infection in an endemic setting (1:160) [44]. Another study conducted from 2013 to 2014 in 100 hospital patients with non-malarial febrile syndrome from Meta Department in Villavicencio measured in 29% of paired sera a four-fold rise in MAT titer, with Canicola and Ballum being the most prevalent serogroups [45]. In Villeta, of 104 patients with UTFI, 24% were seropositive to *Leptospira* spp. (2011–2013) [46], however again with a very low MAT titre cut-off (1:20), and the article did not show the effective measured titres of each sample. A seroprevalence study from 2001–2002 in the Embera-Katio indigenous community found that the seropositivity to *Leptospira* spp. by MAT (cut-off not specified) was 18% [47]. Another seroprevalence study in 353 inhabitants of an urban district of Cali found a seroprevalence of 12% (MAT cut-off titer of $\geq$ 1:100)[48]. Despite this evidence of leptospirosis in other study areas and populations in Colombia and the favorable conditions for the development of leptospirosis in the Wiwa population, based on the results obtained in this study, the Wiwa do not seem to be highly exposed to *Leptospira* spp.. Nevertheless, efforts to further shed light into the leptospirosis burden in these communities should be pursued and testing for leptospirosis in acute febrile illness patients in (sub)tropical regions remains highly recommended.

## 4.1 Limitations

It is well-known that the MAT is an imperfect reference test with a sensitivity and specificity for acute serum of 55% and 97% for subclinical cases and 66% and 98% for clinical cases [49] and 88% and 98% respectively in convalescent serum [23]. Therefore, some of the ELISA false positives results may have actually been true positives (but not recognized by MAT). Hence, the low specificity of the ELISA in cross-sectional sera might be better than described here.

Ideally the samples would have been tested by another serological test and the ELISA then assessed by Bayesian latent class analysis, a method used when assessing the performance of a test with imperfect reference tests [25,50]. However, this was not possible due to budget limitations. In general, testing several available serological tests and identifying the most suitable one for a specific region, would be the ideal approach, but is usually not done due to budgetary reasons.

On the basis of previous ELISA test results at the medical diagnostic laboratory at the Swiss TPH [42], false negative results were not expected due to the high sensitivity of the ELISA, and therefore, not all ELISA-negative results were tested by MAT. Nevertheless, we counted them as true negatives in our seroprevalence estimation of the Colombian study population. Since all ELISA-negative sera, which were tested by the MAT were negative, we are confident of not having introduced error into our seroprevalence estimation. If only the 156 samples tested by the MAT would have been used to estimate seroprevalence, a selection bias would have been introduced (as we chose all positive ELISA results but only a random sample of the negative to be tested by the MAT).

The blood donor sera and the sera of patients with other infections but no diagnosis of leptospirosis were not tested by MAT due to budgetary reasons (with the exception of 3 samples). Nevertheless, we deem the sera suitable to be included in the ELISA evaluation as *Leptospira* spp. negative samples, as a recent clinical *Leptospira* spp. infection can most likely be ruled out. Because blood donors are healthy, leptospirosis is not endemic in Switzerland and patients with other infections than leptospirosis would have been tested for leptospirosis at the Swiss TPH if the disease had been suspected. Nevertheless, a low *Leptospira* spp. antibody titer cannot be ruled out 100%, given the lack of information on travel history from blood donors and a known travel history in some of the patients with other infections.

We generally associate IgM antibodies with acute disease and do not use them in diagnostics in cross-sectional studies in healthy populations. However, IgG antibody production is not always induced with a *Leptospira* spp. infection and IgM antibodies may last for several months to years [5]. Therefore, an IgM detection test can be useful in screening for *Leptospira* exposure in the recent past.

## 5. Conclusions

The Serion IgM ELISA is suitable to diagnose clinical leptospirosis in non-endemic settings such as in acutely ill Europeans several days after onset of disease, nevertheless a confirmatory test is always recommended. However, for cross-sectional studies, the ELISA is not recommended, due to its low specificity. Despite the evidence of a high prevalence of leptospirosis in other study areas and populations in Colombia and the favorable conditions for the development of leptospirosis in the Wiwa population, based on the results obtained in this study, the Wiwa do not seem to be highly exposed to *Leptospira* spp.. Nevertheless, leptospirosis should be considered and tested in patients presenting with febrile illness.

## Informed consent statement

Written informed consent was obtained by each participant/patient/parent/legal guardian before participation in the study implemented in Colombia.

## Supporting information

**S1 File Appendix A—Table with characterization of admission and follow-up serum samples from leptospirosis patients Appendix B—R code for sample size calculation.**
(DOCX)

## Acknowledgments

We would like to thank Dr. Klaus Reither for his constant support, Dr. Tracy Glass for statistical advice (both Swiss TPH) and Dr. Alba Luz-Luque Lommel for her support in Colombia. We also thank Professor Mark Stevenson (University of Melbourne) for providing the R code for the sample size calculation.

## Author Contributions

**Conceptualization:** Anou Dreyfus, Marie-Thérèse Ruf, Sven Poppert, Gustavo Concha, Jorge Orozco, Johana Colorado, Juan C. Dib, Simone Kann.

**Data curation:** Anou Dreyfus, Marie-Thérèse Ruf.

**Formal analysis:** Anou Dreyfus, Eliharintsoa Rajaonarimirana.

**Funding acquisition:** Anou Dreyfus, Sven Poppert, Daniel H. Paris, Simone Kann.

**Investigation:** Anou Dreyfus, Marie-Thérèse Ruf, Marga Goris, Anne Mayer-Scholl, Nadine Loosli, Nadja S. Bier, Gustavo Concha, Johana Colorado, Andrés Aristizábal.

**Methodology:** Anou Dreyfus, Marie-Thérèse Ruf.

**Project administration:** Anou Dreyfus, Anne Mayer-Scholl, Nadja S. Bier, Simone Kann.

**Resources:** Marie-Thérèse Ruf, Daniel H. Paris, John Stenos, Simone Kann.

**Supervision:** Anou Dreyfus, Marie-Thérèse Ruf.

**Validation:** Anou Dreyfus, Marie-Thérèse Ruf.

**Visualization:** Anou Dreyfus.

**Writing – original draft:** Anou Dreyfus.

**Writing – review & editing:** Anou Dreyfus, Marga Goris, Sven Poppert, Anne Mayer-Scholl, Nadja S. Bier, Daniel H. Paris, Tshokey Tshokey, John Stenos, Simone Kann.

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
