## [Decision Letter · Decision Letter 0]

19 Jan 2022

Dear Dr. Dreyfus,

Thank you very much for submitting your manuscript "Leptospira spp. seroprevalence in the Wiwa indigenous population from Colombia and the comparison of the Serion IgM ELISA and Microscopic Agglutination Test for diagnosis of Leptospira spp. infections in human sera from different geographical origins" for consideration at PLOS Neglected Tropical Diseases. As with all papers reviewed by the journal, your manuscript was reviewed by members of the editorial board and by several independent reviewers. In light of the reviews (below this email), we would like to invite the resubmission of a significantly-revised version that takes into account the reviewers' comments. 

As some reviewers pointed out, the objectives, methods and analysis needs to be consistent and aligned to answer the research question. This manuscript's objective is to evaluate the performance of the Serion ELISA, therefore, sample size and methods need to focused on that objective. As diagnostic tests need to be evaluated for their specific use, clearly separate the evaluation for i) clinical diagnosis and for ii) prior exposure detection. 

For each one these sub-objectives, method and results need to reflect the standard practices for reporting diagnostic test evaluation:

https://journals.plos.org/plosone/s/best-practices-in-research-reporting

https://www.equator-network.org/reporting-guidelines/stard/

Additionally, considering the small sample size of the evaluation for the clinical diagnosis, more details regarding its contribution and comparison with several others papers evaluating the same kit are needed under the rationale for the study.

The sub-objective evaluating the accuracy to detect prior exposure needs a better description of the design and analytical approach as MAT is a reference test but not a perfect "gold" standard and results correspond to relative SE and SP. Authors can consider analytical methods for evaluation of diagnostic tests with no gold standard and two populations. The effective sample size are the number of samples with both ELISA and MAT results. 

The goal of estimating sero-prevalence in the Colombian study population is a secondary outcome of the accuracy study, and stated, most of the report should be focused on the diagnostic test evaluation study following recommended guidelines.

We cannot make any decision about publication until we have seen the revised manuscript and your response to the reviewers' comments. Your revised manuscript is also likely to be sent to reviewers for further evaluation.

Sincerely,

Claudia Munoz-Zanzi

Associate Editor

Elsio Wunder Jr

Deputy Editor

As some reviewers pointed out, the objectives, methods and analysis needs to be consistent and aligned to answer the research question. This manuscript's objective is to evaluate the performance of the Serion ELISA, therefore, sample size and methods need to focused on that objective. As diagnostic tests need to be evaluated for their specific use, clearly separate the evaluation for i) clinical diagnosis and for ii) prior exposure detection. 

For each one these sub-objectives, method and results need to reflect the standard practices for reporting diagnostic test evaluation:

https://journals.plos.org/plosone/s/best-practices-in-research-reporting

https://www.equator-network.org/reporting-guidelines/stard/

Additionally, considering the small sample size of the evaluation for the clinical diagnosis, more details regarding its contribution and comparison with several others papers evaluating the same kit are needed under the rationale for the study.

The sub-objective evaluating the accuracy to detect prior exposure needs a better description of the design and analytical approach as MAT is a reference test but not a perfect "gold" standard and results correspond to relative SE and SP. Authors can consider analytical methods for evaluation of diagnostic tests with no gold standard and two populations. The effective sample size are the number of samples with both ELISA and MAT results. 

The goal of estimating sero-prevalence in the Colombian study population is a secondary outcome of the accuracy study, and stated, most of the report should be focused on the diagnostic test evaluation study following recommended guidelines.

Reviewer's Responses to Questions

**Key Review Criteria Required for Acceptance?**

**Methods**

-Are the objectives of the study clearly articulated with a clear testable hypothesis stated?

-Is the study design appropriate to address the stated objectives?

-Is the population clearly described and appropriate for the hypothesis being tested?

-Is the sample size sufficient to ensure adequate power to address the hypothesis being tested?

-Were correct statistical analysis used to support conclusions?

-Are there concerns about ethical or regulatory requirements being met?

Reviewer #1: The authors described the study objectives clearly. The study design, including sample size, population sources, and statistical analysis, were explained well. However, the approvals of the Human Research Ethics committee on using serum samples from Group A and B are not described. And I need a rationale on why the study using Bhutanese specimens was granted by the Human Research Ethics committee of the University of Newcastle, Australia, only.

Reviewer #2: Introduction

1.- Objectives can be clearly summarized and stated. Lines 148-162 may be better included in the materials and methods

Material and methods

2.- Line 181, indicate in the first time what “TPH” means

3.- Line 295. What is an ambiguous result? Why an ambiguous result is considered negative? There were duplicates that could be made to disambiguate

Reviewer #3: inclusion of all groups is not clear according with objetive of study: compare two diagnoses test and the seroprevalence of Wiwa population. 

Two different topics. Sample size should be different: test for a test or population survey. 

Ethical OK

**Results**

-Does the analysis presented match the analysis plan?

-Are the results clearly and completely presented?

-Are the figures (Tables, Images) of sufficient quality for clarity?

Reviewer #1: The authors demonstrate results and the data analysis pretty clearly and entirely as the study objectives. The Figures and Tables are tidy and easy to read. In Table 4, the "N" for Group E should be 156 instead of 321. To fulfil more on the study objectives, the accuracy of the "Serion ELISA classic Leptospira IgM" test inferring seroprevalence in sera Group D and E--the endemic areas. Additional positive predictive value (PPV) and negative predictive value (NPV) of the ELISA test in Table 4 (MAT used the reference) will directly point out the failure of using ELISA to investigate the seroprevalence in the endemic areas.

Reviewer #2: results

4.- In the table 4, groups A + B should have a single value of "N" (34) to maintain congruence with the other data in the table.

5.- In group C, why can't the 95% CI for specificity be calculated?

6.- In group E, the reference test was carried out in 156 people and not in 321.

7.-Line 359. In this section, authors should not incorporate sensitivity in the interpretation, because in group C only specificity is evaluated.

8.-Figure 1. Were some of the random samples considered MAT (+)? If any were, it would not be correct to include all the "ELISA negative" (476 or 163) in the sero-negative group, because in theory the ELISA-negative have 2X or 5X with respect to the random sample and if they have some MAT + in the random sample group, there would also be some MAT + in the "ELISA negative" group. With this, the sero-prevalence could be re-calculated.

9.-The paragraph on lines 441-451 corresponds to the limitations of the study. They must be sent to the corresponding section.

Reviewer #3: Table 4 should include total, positives by MAT and ELISA.

the use of the term seroprevalence of the study is not clear because are different populations, different countries and sample size, not methodological correct.

**Conclusions**

-Are the conclusions supported by the data presented?

-Are the limitations of analysis clearly described?

-Do the authors discuss how these data can be helpful to advance our understanding of the topic under study?

-Is public health relevance addressed?

Reviewer #1: The authors conclude the study reasonably, including all limitations of the analysis and study design.

Reviewer #2: 9.- The first paragraph (lines 416-424) is part of the objectives and methodology that has already been explained. It is recommended to start the discussion with the main results obtained.

10.- Lines 458-479 support the over and underestimation assumption. Therefore, they must be linked to the sentence that mentions it.

11.-Line 494. Has the hypothesis of low specificity been evidenced in other studies of leptospira or other spirochetes? If so, include those studies as references.

Reviewer #3: It is inconsistent with the introduction the profile of risk of Wiwa population and the conclusion presented. 

The statement "ELISA is technically working well..." should include the concrete arguments for that.

The discrepancy between MAT and ELISA should be analice from the theory more than make comparisons with results from other species.

conclusions about seroprevalence in Wiwa population should be discuss and evaluate in a future with comorbilities from the region as other febrile syndromes from infectious diseases.

**Editorial and Data Presentation Modifications?**

Reviewer #1: I suggest a Minor Revision for this manuscript. My significant comments are as follows:

1. show IRB approval for study Group A & B and IRB multi-centre or Bhutanese IRB approval for study Group D 

2. correct the "N" number in Table 4 for Group E

3. add PPV (95%CI) and NPV (95%PI) in Table 4 for Group D and E.

Reviewer #2: Be congruent with the number of decimal places to include in the percentages.

In Abstract: Maintain the same units at the conclusion of the% seroprevalence (line 59)

Reviewer #3: Maybe a figure (a different figure than a diagram) to show some results will be more easy to understand (388-399).

**Summary and General Comments**

Reviewer #1: Some significant results were published in Pathogens; the whole story is still beneficial for Leptospira diagnosis or seroprevalence study. It emphasizes the essential test validation before use in disease-endemic areas.

Reviewer #2: This study has the strength of having investigated the performance of a serological test for leptospirosis in a non-endemic setting (after onset of disease) and in an endemic setting where further research is still necessary.

Reviewer #3: I think that more importance should be done to wiwa results. And as secondary results should be comparisons with other populations. That implies more emphases at those results and to begin with Wiwa populations context, methodology and results.

PLOS authors have the option to publish the peer review history of their article (what does this mean?). If published, this will include your full peer review and any attached files.

Reviewer #1: No

Reviewer #2: Yes: Antonio Flores

Reviewer #3: Yes: Janeth Perez-Garcia
---

## [Decision Letter · Decision Letter 1]

26 Mar 2022

Dear Dr. Dreyfus,

Thank you very much for submitting your manuscript "Comparison of the Serion IgM ELISA and Microscopic Agglutination Test for diagnosis of Leptospira spp. infections in sera from different geographical origins and estimation of Leptospira seroprevalence in the Wiwa indigenous population from Colombia" for consideration at PLOS Neglected Tropical Diseases. As with all papers reviewed by the journal, your manuscript was reviewed by members of the editorial board and by several independent reviewers. The reviewers appreciated the attention to an important topic. Based on the reviews, we are likely to accept this manuscript for publication, providing that you modify the manuscript according to the review recommendations. 

Sincerely,

Elsio Wunder Jr, D.V.M., Ph.D.

Deputy Editor

Reviewer's Responses to Questions

**Key Review Criteria Required for Acceptance?**

**Methods**

-Are the objectives of the study clearly articulated with a clear testable hypothesis stated?

-Is the study design appropriate to address the stated objectives?

-Is the population clearly described and appropriate for the hypothesis being tested?

-Is the sample size sufficient to ensure adequate power to address the hypothesis being tested?

-Were correct statistical analysis used to support conclusions?

-Are there concerns about ethical or regulatory requirements being met?

Reviewer #1: Authors describe the objectives of each experiment clearly. The primary aims are evaluating the analysis sensitivity and specificity of the Serion ELISA classic Leptospira IgM using a) 14 confirmed leptospirosis cases originated from Netherland, b) 20 healthy blood donors, c) 41 confirmed other infections originated from Switzerland, d) 321/489 sera from volunteers (healthy or ill) originated from Colombia. 

The objectives of the group d investigation (line 191-193) are arguable. In basic immunology, IgG antibody screenings are usually performed for investigating seroprevalence. The IgM ELISA was used for seroprevalence of leptospirosis in this study. Authors should describe reasons for use this unexpected study design. Previous studies on validating IgM and IgG for Leptospira seroprevalence in comparison are recommended in the Introduction section. Emphasizing using MAT on single sera for the seroprevalence (not diagnosis) as a reference method should be done to remind readers. 

The study design of the seroprevalence in group d is questionable. There is a cross-sectional study design that perform a reference test first and then performing case-control sampling (based on the reference test results) to further perform index test (the ELISA test in this study). Authors should describe the type of the cross-sectional designs selected in this study and the reason of selections. 

PPV and NPV are not needed for group a, b or c. 

In my personal opinion, all a,b, c and d should be done for only one primary aim: to investigating the accuracy of the ELISA test in survey sero-prevalence (both past and current infection) compared to MAT using single sera collected during leptospiremia, leptospirurea, and no Leptospira in bodies.

Reviewer #2: objectives.

Please to include the text suggested for the author: 

"“The objectives of our study were to evaluate the sensitivity and specificity of the commercially

available Serion ELISA classic Leptospira IgM using the MAT as imperfect reference test with patient

sera from confirmed leptospirosis cases, blood donor sera, with sera from patients with other

infections than leptospirosis (all European) and sera from a cross-sectional study in the indigenous

population called “Wiwas” of the Sierra Nevada de Santa Marta, North-east of Colombia..."

Reviewer #3: (No Response)

**Results**

-Does the analysis presented match the analysis plan?

-Are the results clearly and completely presented?

-Are the figures (Tables, Images) of sufficient quality for clarity?

Reviewer #1: Figure 1--It is not correct to pool ELISA and MAT results (called sero-positive and sero-negative). 

Table 4 --the 124/32 (instead of NA) should be add in the ELISA (pos/neg) column in row D ii

Estimating sero-prevalence based on the ELISA in Wiwa population were less accuracy as the significant low specificity compared to those estimated by MAT. Combination results from samples having ELISA results only might be another kind of bias.

Reviewer #2: Table 4.

specificity values of Group B are incorrectly added within the group A row. (Group A are cases, there are no "negative samples", so it should not have any specificity value)

In group C, please indicate what the value of 100% specificity corresponds to. At 3 MAT-/ELISA- ? if so, clarify it in the legend below

Reviewer #3: (No Response)

**Conclusions**

-Are the conclusions supported by the data presented?

-Are the limitations of analysis clearly described?

-Do the authors discuss how these data can be helpful to advance our understanding of the topic under study?

-Is public health relevance addressed?

Reviewer #1: Authors conclude that the IgM ELISA test was less accuracy when use for Leptospira seroprevalence compared to MAT. Authors argue on the less sensitivity of MAT (single sera) when use as a diagnosis tool (line 479-483). However, I did not see the relevant to the group d experiment. 

Even the results clearly show low accuracy of the IgM ELISA test, authors insist that ".. IgM detection test can be useful in screening for Leptospira exposure in the recent past.." (510-517) based on a citation of book reference. Serious discussion with previous reports on this issue are lacking.

Reviewer #2: Discussion: About your hypothesis in lines 422-424, It is Ok after your feedback.

Reviewer #3: (No Response)

**Editorial and Data Presentation Modifications?**

Reviewer #1: (No Response)

Reviewer #2: none

Reviewer #3: 5 In the title could be more accurate to use leptospirosis seroprevalence 

51 Sensitivity value at cross sectional study in Colombia is missing in the abstract

184 “most likely seronegative for Leptospira spp..” could means doubt about leptospirosis diagnoses and cannot be objective to evaluate cross reaction. 

254 Should be interesting pointed the reason to use OIE standards for human test, and not WHO.

278 If you referred to “endemic places” to Colombia, should be better to describe it explicitly, because its methodology of the study.

**Summary and General Comments**

Reviewer #1: I concerns on the objectives and study designs used in this study. It might be better that authors present a major and accuracy key message to readers with evidence based style. The manuscripts can be confusing as the ELISA was evaluated as a diagnosis tool and a sero-prevalence in the same article. As a reader of PLoS NTD, I eager to know if IgM ELISA (using non-pathogen Leptospira antigen) is useful for seroprevalence in the low income countries with leptospirosis endemic areas or not.

I am confused by Abstract. And please check the information accuracy in the abstract as well.

Reviewer #2: none

Reviewer #3: (No Response)

PLOS authors have the option to publish the peer review history of their article (what does this mean?). If published, this will include your full peer review and any attached files.

Reviewer #1: No

Reviewer #2: Yes: Antonio Flores

Reviewer #3: Yes: Janeth Perez-Garcia

Figure Files:

Data Requirements:

Reproducibility:

References

---

## [Editor Report · Decision Letter 2]

22 Apr 2022

Dear Dr. Dreyfus,

We are pleased to inform you that your manuscript 'Comparison of the Serion IgM ELISA and Microscopic Agglutination Test for diagnosis of *Leptospira* spp. infections in sera from different geographical origins and estimation of *Leptospira* seroprevalence in the Wiwa indigenous population from Colombia' has been provisionally accepted for publication in PLOS Neglected Tropical Diseases.

There are one minor suggestion for change in the text. On page 80, the authors mention that the *Leptospira* genus has 64 species. A recent publication (PMID: 34914572) has identified 4 new species for the genus, bringing the total to 68.

Best regards,

Elsio Wunder Jr, D.V.M., Ph.D.

Deputy Editor

---

## [Editor Report · Acceptance letter]

30 May 2022

Dear Dr. Dreyfus,

We are delighted to inform you that your manuscript, "Comparison of the Serion IgM ELISA and Microscopic Agglutination Test for diagnosis of Leptospira spp. infections in sera from different geographical origins and estimation of Leptospira seroprevalence in the Wiwa indigenous population from Colombia," has been formally accepted for publication in PLOS Neglected Tropical Diseases.

Best regards,

Shaden Kamhawi

co-Editor-in-Chief

Paul Brindley

co-Editor-in-Chief
